# Recurrence of Upper Extremity Deep Vein Thrombosis Secondary to COVID-19

**DOI:** 10.3390/v13050878

**Published:** 2021-05-11

**Authors:** Yesha H. Parekh, Nicole J. Altomare, Erin P. McDonnell, Martin J. Blaser, Payal D. Parikh

**Affiliations:** 1Department of Medicine, Rutgers Robert Wood Johnson Medical School, New Brunswick, NJ 08901, USA; yhp7@rwjms.rutgers.edu (Y.H.P.); nja71@rwjms.rutgers.edu (N.J.A.); em912@rwjms.rutgers.edu (E.P.M.); 2Center for Advanced Biotechnology and Medicine, Rutgers University, New Brunswick, NJ 08901, USA; 3Department of Medicine, Rutgers Robert Wood Johnson Medical School, Piscataway, NJ 08851, USA

**Keywords:** SARS-CoV-2, COVID-19-associated coagulopathy, coagulation disorders, venous thromboembolism

## Abstract

Infection with SARS-CoV-2 leading to COVID-19 induces hyperinflammatory and hypercoagulable states, resulting in arterial and venous thromboembolic events. Deep vein thrombosis (DVT) has been well reported in COVID-19 patients. While most DVTs occur in a lower extremity, involvement of the upper extremity is uncommon. In this report, we describe the first reported patient with an upper extremity DVT recurrence secondary to COVID-19 infection.

## 1. Introduction

The COVID-19 pandemic, due to severe acute respiratory syndrome-coronavirus-2 (SARS-CoV-2) has had worldwide consequences [1]. The clinical and pathological features of the infection are gradually becoming better understood [2]. While COVID-19 infects the respiratory tract and presents as pneumonia in most patients, others also suffer from severe neurological, cardiovascular, and/or gastrointestinal complications due to hyperinflammatory and hypercoagulable states [3,4,5,6]. Coagulation disorders have been noted in 23–49% of COVID-19 patients regardless of the use of heparin or low-molecular-weight heparin [7]. Arterial and venous thromboembolic events are more common than bleeding disorders, with the highest rates described in ICU settings [8,9,10,11,12]. COVID-19 hospitalized patients have had much higher DVT rates (31% in one New York study) than in hospitalized patients without COVID-19 (19%) [13]. 

Most DVTs occur in the lower extremities (LEDVT), due to increased gravitational stress and decreased endothelial fibrinolytic activity compared to upper extremity veins [14]. Only 4–10% of DVTs occur in the upper extremities (UEDVT), which may be primary (20%) or secondary (80%) [15,16,17]. Primary UEDVTs, identified as two events per 100,000 patients, are either idiopathic or due to effort-induced injuries or anatomical variation, like Paget–Schroetter syndrome or cervical rib [16,17,18,19]. Secondary UEDVTs may be caused by malignancy or, more commonly, by intravenous catheters or pacemaker wires, especially after their insertion [16,17,18,19,20]. 

Although UEDVT incidence is low, severe complications include pulmonary embolism, post-thrombotic syndrome, and death. Pulmonary embolism (PE) may occur in up to 30% of patients following a UEDVT and may be fatal [15,17]. The mortality rate in patients with UEDVT varies from 15–50%, chiefly related to underlying conditions including malignancy, infection, or organ failure [15,17]. Another disabling complication, post-thrombotic syndrome, presents as swelling, pain, and limb fatigue with exertion in 27–50% of UEDVT patients [17]. UEDVTs recur in 9% of patients, usually in the ipsilateral extremity, whereas LEDVT recurrence may reach 97% [16,21].

While both UEDVTs and LEDVTs secondary to COVID-19 infection have been reported [22,23], there is little information on COVID-19 as a risk factor for recurrent UEDVT and how to optimize management. We present the case of an 85-year-old patient, who to our knowledge, was the first to develop recurrent UEDVT as the presenting sign of an asymptomatic COVID-19 infection.

## 2. Materials and Methods

Written consent by the patient was obtained to receive medical records from the 2017 and 2020 hospitalizations for UEDVT.

## 3. Case Presentation

At age 81, the patient was diagnosed with his first episode of UEDVT in 2017. His past medical history is significant for long-standing hypertension, hypercholesterolemia, type 2 diabetes mellitus, and coronary artery disease post-myocardial infarction with residual severe ischemic cardiomyopathy. He has no prior personal or family history of clotting or bleeding disorders. He has a 15-pack-year tobacco smoking history, quitting at age 29, and denies illicit drug use. Until his cardiac symptoms worsened, he had an active lifestyle of bicycling and playing tennis weekly. 

In November 2016, he underwent a biventricular pacemaker upgrade without significant post-procedure complications. Following the pacemaker manipulation, he was instructed to have limited upper extremity movement for 6 weeks, and then resumed playing tennis. While playing tennis, 42 days post-procedure, the patient noted new, left upper extremity swelling from the top of his wrist to his shoulder, which lasted for 4 days. Upon presentation to the ER, on physical exam he had 2+ nonpitting edema from the left hand to the upper arm with mild tenderness, but without phlegmasia or erythema. His right upper extremity had no edema but he had trace nonpitting edema around the ankles bilaterally with prominent venous pattern. He was not having acute distress or labored respiration on room air and his chest was clear to auscultation bilaterally. Portable chest X-ray revealed cardiomegaly with vascular congestion and interstitial densities. On admission, laboratory results (Table 1) were significant for normal PT, INR, and aPTT, but decreased platelet and white and red blood cell counts. Venous duplex ultrasound showed left axillary and brachial vein UEDVTs, but the left subclavian vein could not be adequately visualized due to the presence of a left-sided pacemaker. The UEDVT was considered secondary to subclavian vein trauma from the left-sided pacemaker placement 42 days previously.

To treat this DVT, he received one dose of enoxaparin in the emergency department. On hospital admission day 2, his medical team continued his enoxaparin and home aspirin (325 mg daily) but stopped his home clopidogrel (75 mg daily). Interventional radiology consultation recommended anticoagulation treatment and monitoring of the symptoms, without the need for urgent thrombolysis or thrombectomy. He was transitioned to apixaban (5 mg BID) on hospital admission day three, which he then took for the next 8 months. He ultimately underwent a thrombectomy and angioplasty on 13 February 2017. Following the anticoagulation regimen, the patient’s left arm returned to normal and he did not experience any residual symptoms or complications. He remained physically active and continued playing tennis three times a week.

In November 2020, 4 years after his prior episode, the patient sought attention from his primary care physician for new, left arm swelling. He had noted the swelling for 7 days, without pain or erythema, shortness of breath, or chest pain. His home medications were significant for 81 mg of aspirin daily. Venous duplex ultrasonography showed left proximal and mid-brachial vein UEDVT. He was admitted to the hospital for further management. On physical exam, he had left forearm and hand edema that did not extend into his upper arm. Superficial veins were seen in the supraclavicular and infraclavicular region with normal left arm motor and sensation function. Right upper extremity and bilateral lower extremities were without edema. On admission, laboratory results (Table 1) showed slightly elevated PT, INR, and aPTT but decreased platelet and white and red blood cell counts. As part of routine hospital protocol for all admitted patients, he was tested for SARS CoV-2 by PCR of a nasopharyngeal PCR swab.

He was diagnosed with recurrent UEDVT, and a vascular surgery consultant recommended anticoagulation via heparin infusion and left arm elevation with ACETM bandage wrapping since the patient’s central veins were patent. He was admitted to the hospital and discharged the next day to receive apixaban, 10 mg BID for 1 week, followed by 5 mg BID lifelong treatment barring any bleeding episodes. His COVID-19 test was found to be positive after discharge from the hospital and he self-quarantined for 10 days. Aside from the UEDVT, the patient was asymptomatic for COVID-19. While his first UEDVT episode in 2017 was uncomplicated with no sequela, this recurrent UEDVT event was complicated by continued post-thrombotic syndrome, with limited ability to flex at the elbow, 5 months after the UEDVT recurrence.

## 4. Discussion

UEDVTs are uncommon, accounting for 4–10% of all venous thromboembolic events (VTE); 5–42% of UEDVTs occur in the axillary veins compared to the 4–13% in brachial veins [17]. According to a study by Joffe et al., insertion of a central venous catheter was associated with a 7-fold increase in the odds of developing a UEDVT [24]. The risk factors associated with UEDVT and LEDVT were not the same, providing a rationale for differential treatment and prophylaxis regimens. Of patients with pacemaker or internal cardiac defibrillator insertion who developed thrombosis, 59% of thrombotic events occurred within 3 months post-implantation [25]. Our patient’s left axillary and brachial vein UEDVT in 2017 occurred within 42 days after his left-sided pacemaker implantation. Prolonged stasis after pacemaker implantation and effort-induced tennis injury likely also contributed to the thrombotic process. Since the patient’s pacemaker provides a persistent risk factor for UEDVT [25], he would have benefited from long-term, oral anticoagulant treatment after the 8-month course of apixaban following his first UEDVT episode. 

Only 2.4% of patients with a history of UEDVT, who account for 4–10% of all VTEs, have a recurring episode, predisposing them to future recurrences; recurrent UEDVT is an extremely uncommon event [16,17,26,27]. Furthermore, there have been no prior reports of UEDVT in an asymptomatic COVID-19 patient. Most patients develop a UEDVT recurrence in their ipsilateral arm, as did our patient [27]. The strongest risk factors for UEDVT recurrence include thrombophilia due to genetic mutations, including factor V Leiden, protein C, protein S deficiency, and hyperhomocysteinemia, as well as strenuous upper extremity muscle effort [27]. Our patient has not been tested for genetic mutations in his coagulation cascade but was an avid tennis player throughout his life, until age 83, 2 years prior to his recurrence, making mutation unlikely to be the etiology of UEDVT recurrence. 

Of 115 UEDVT patients studied by Martinelli et al., after 6 months of oral anticoagulation therapy or 3 months of either subcutaneous heparin or antiplatelet agents, 34% had a residual vein thrombosis [27]. The authors further reported that the rates of developing UEDVT recurrence in patients with incomplete recanalization and those with complete recanalization were not significantly different [27]. One study demonstrated the rate of recanalization of thrombosed vein of the upper limb using a catheter-directed thrombolysis (CDT) or pharmacomechanical thrombolysis (PMT) using a phlebography or ultrasound immediately after and 1 year after the procedure [28]. The CDT group’s immediate success rate was 91.7% compared to 100% in the PMT group, while the 1-year vein patency rate was 91.7% in the CDT group versus 94.7% in the PMT group [28]. Since our patient underwent a thrombectomy and angioplasty of his UEDVT 1 month after his diagnosis, it is less likely that his UEDVT recurrence 4 years later was due to incomplete recanalization of the left axillary and brachial vein.

An important area of research has been the relationship between COVID-19 and thrombosis. The risk of thrombosis reflects Virchow triad of endothelial injury, stasis, and a hypercoagulable state [29]. Endothelial injury occurs when the virus invades endothelial cells via ACE2 receptors, activating the renin-angiotensin system, increasing angiotensin II levels [29]. This induces expression of tissue factor and plasminogen activator inhibitor 1, favoring coagulation [29,30]. Endothelial injury is enhanced by release of inflammatory cytokines, acute phase reactants, and complement pathway activation, as well as intravascular catheter insertion [29]. Finally, the COVID-19-induced hypercoagulable state, due to increased early thrombin burst, increased fibrin generation, greater clot strength, and reduced fibrinolysis, following elevated levels of prothrombotic factors like D-dimer, fibrinogen, factor VIII, and anionic phospholipids [29]. Because of the increased thrombosis risk, COVID-19 patients should have complete blood count (CBC), PT, aPTT, fibrinogen, and D-dimer assays obtained on hospital admission [29].

Although our patient did not have a D-dimer test, his laboratory values were significant for slightly elevated PT, INR, and aPTT during the UEDVT recurrence in 2020 (Table 1), despite their normal levels during the 2017 UEDVT event. The patient also had anemia, thrombocytopenia, and neutropenia during both UEDVT episodes. He has a history of thrombocytopenia since 2011. Further workup in 2017 revealed elevated immature platelet fraction indicating peripheral platelet destruction. We presume that the hypercoagulable state of COVID-19 led to an UEDVT recurrence despite thrombocytopenia. 

The incidence of LEDVT in hospitalized COVID-19 patients also has been an area of increased concern. Among COVID-19 patients in Wuhan, China, the incidence of LEDVT was 35.2%, of whom 90% were admitted to the intensive care unit (ICU), indicating severity of illness as an important correlate [31]. In another Wuhan study, compared to non-LEDVT patients, LEDVT patients were more likely to be >65 years old and with lower oxygenation status, increased leg pain, and greater extent of stasis due to being bed-ridden and COVID-19 illness severity [32]. Among non-ICU COVID-19 patients in Rome, Italy, the incidence of LEDVT diagnosed by venous compression ultrasonography ~6 days after hospitalization was 11.9% [33]. This demonstrates that the hypercoagulable state induced by COVID-19 can occur despite thromboprophylaxis, with COVID-19 serving as an independent risk factor for DVT in hospitalized patients [32,33]. There was no significant difference among patients with DVT and those without, with regard to patient comorbidities, including obesity, hypertension, type 2 diabetes mellitus, and coronary artery disease, or demographic factors like age or sex [33]. Avruscio et al. found that DVTs occurred in 42.4% of hospitalized patients with COVID-19, with 25% of those DVTs occurring in the upper extremities. The risk of VTE was five times higher in patients with COVID-19 than those without [34]. All the COVID-19 patients received prophylactic or high-dose anti-coagulation. This patient did not receive the anti-coagulation expected for someone with his health history and COVID-19 infection, perhaps because he was otherwise asymptomatic. 

LEDVTs recur ~20% of the time [35]. Risk factors for recurrence include smoking, age >65 years old, idiopathic initial LEDVT, discontinuation of anticoagulation therapy, and patient comorbidities, like diabetes mellitus, dyslipidemia, obesity, and malignancy [35,36]. Ipsilateral recurrence is associated with increased risk of post-thrombotic syndrome [36]. While the incidence of LEDVT recurrence secondary to COVID-19 is unknown, in one prospective Spanish study examining long-term (>90 days) outcomes in COVID-19 patients diagnosed with VTE, there were no VTE recurrences [37]. 

Only a few UEDVTs secondary to COVID-19 infection have been reported [21,22,34]. Patients with COVID-19 who have illness sufficiently severe to require continuous positive airway pressure therapy can have induction of UEDVT by compression of the superficial or deep veins of the upper extremities [21,22]. In another study determining the incidence of VTE in COVID-19 patients admitted to the medical wards versus the ICU, all 9 patients who had a UEDVT were admitted to the ICU [34]. Unlike those patients, our patient was found to be COVID-19 positive after seeking medical attention for his left arm swelling. That our patient did not need supplemental oxygen during his overnight hospital stay demonstrates that patients may have increased thrombosis risk regardless of COVID-19 illness severity. Furthermore, COVID-19 patients who are otherwise asymptomatic who present with VTE may represent a coagulation diathesis that differs from patients with increased disease severity requiring supplemental oxygenation or ICU level of care. 

The current guidelines for UEDVT treatment parallel the initial treatment protocol for LEDVT, which includes unfractionated heparin (UFH) or low-molecular-weight heparin (LMWH) followed by 3 to 6 months of maintenance therapy [14]. Direct oral anticoagulants (DOAC) like rivaroxaban or apixaban, which were prescribed for this patient, can also be used for initial treatment [14]. While prophylaxis of LEDVT recurrence is usually highly effective, UEDVT recurrences are not easily prevented [14]. While there are no published recommendations for UEDVT secondary to COVID-19, for COVID-19 patients diagnosed with proximal DVT or PE a minimum of 3-month anticoagulation therapy has been recommended [35]. Moores et al. recommend the standard dose of anticoagulant thromboprophylaxis given parentally for acutely ill patients admitted to the hospital, and mechanical prophylaxis as well. For outpatient COVID-positive patients, the authors recommend apixaban, dabigatran, rivaroxaban or edoxaban therapy for at least 3 months [38]. In COVID-19 patients with recurrent VTE despite anticoagulation therapy, increasing the weight-adjusted LMWH dose by 25–30% appears beneficial [35]. Further investigation into specific treatment and prophylaxis protocols for UEDVT, as well as UEDVT secondary to COVID-19 infection, is needed.

## 5. Conclusions

We described the first case of recurrent UEDVT as the presentation of an otherwise asymptomatic SARS-CoV-2 infection. During these pandemic times, we recommend considering SARS-CoV-2 infection as the etiology of venous thromboembolic events whether in the presence or absence of characteristic COVID-19 symptoms. With insufficient knowledge of prophylactic strategies for UEDVT patients, further investigation is needed to determine optimal approaches for treating and preventing UEDVT recurrence during the COVID-19 pandemic and beyond.

## Figures and Tables

**Table 1 viruses-13-00878-t001:** Significant laboratory values from the 2017 and 2020 hospital admissions.

	Reference Values (2017)	1 January 2017	Reference Values (2020)	24 November 2020
**PT ^1^**	11.7–15 s	14	9.2–11.7 s	12.2
**INR ^2^**	0.81–1.20	1.10	0.88–1.13	1.17
**aPTT ^3^**	22.9–36.3 s	35.4	21.7–30.5 s	30.3
**WBC**	4.5–11 × 10^3^/μL	3.9	4.5–11 × 10^3^/μL	3.7
**RBC**	4.5–5.9 × 10^6^/μL	3.4	4.5–5.9 × 10^6^/μL	3.8
**Hemoglobin**	11.6–16.3 g/dL	10.5	11.6–16.3 g/dL	11.5
**Hematocrit**	40–52%	32.5	40–52%	36.6
**Platelet**	150–400 × 10^3^/μL	99	150–400 × 10^3^/μL	89

^1^ PT = prothrombin time; ^2^ INR = international normalized ratio; ^3^ aPTT = activated partial thromboplastin time.

## Data Availability

No new data were created or analyzed in this study. Data sharing is not applicable to this case report.

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
