# Peer review of "Recurrence of Upper Extremity Deep Vein Thrombosis Secondary to COVID-19"

_viruses, 2021, doi:10.3390/v13050878_

Round 1
Reviewer 1 Report
Many thanks to the Authors that tried to reply to all my comments, but I still disagree with them for the most part of their answers. Namely:
- The text has been modified; they could have added another reference (Avruscio G et Al, Clin Trans Sci 2020), but this could seem fair.
- With such a high CHADS2VASc score (> 3) in my opinion this patient had to mandatorily receive oral anticoagulant treatment (even with aspirin only, rather that DAT), above all after his first UEDVT in 2017, rather than prescribe an 8-months course only of apixaban after stopping clopidogrel
- For the reasons above reported, the patient had to continue oral anticoagulant treatment with apixaban after 8 months, even lowering the dose to 2.5 mg twice-daily, because of the persistence of the risk factor causing the UEDVT represented by the PM
- What about thrombocytopenia? Which was the platelet count during the anticoagulant treatment? Was it transitory and adjustable or chronic? Concerning the data about the recanalization rate I think that there was a misunderstanding: I did not refer to the data of the literature, but at the data of the patient. There is no mention of the ultrasound scan reporting the recanalization rate of the thrombosed veins of the upper limb
- Anyway, irrespective of the recanalization rate, in 2020 the patient experienced a SECOND deep thrombotic event, and for this reason he should have received a long-term course of oral anticoagulant treatment, according to the International Guidelines, and not only for 1 year, since he had no contraindications to receive this therapy.
- It is my opinion that at this very moment when a patient has a suspected DVT, even of the upper limbs, he/she refer immediately to an ER, where currently ALL patients are routinarily tested for SARS-CoV-2 infection before their admission, sinche the most of the patients are asymptomatic and the infection is determined like a screening. In thi way, the recommendation of the Authros that all patients with VTE should be tested for SARS CoV-2 infection....probably is not a recommendation because this behaviour still exists in many ER departments worldwide
- The last issue: I do not understand why, even on anticoagulant treatment for a not life-threatening deep vein thrombosis, the patient underwent thrombectomy and angioplasty of the axillary and subclavian veins. Is it a common procedure in their hospital? Because I think that it is not in many other hospitals around the world
Reviewer 2 Report
The authors have significantly improved the manuscript, which now looks much better, and its substantive value and clinical significance is high.
I am asking the authors to update the literature: L26 (Coagulation disorders have been...) - doi: 10.1055/a-1346-3178.
Reviewer 3 Report
It's a good idea to look at the coagulation disorder in COVID patients, which has been reported often in ICU settings. It's no surprise that COVID patients more likely developed thrombotic events, such as deep vein thrombosis. However, the underlying mechanism still unclear, in which inflammatory response with the monocytes, neutrophils, and macrophages infiltration are recognized.
- It would be great if the authors could give more evidence or explain the underlying association of COIVD infection and UEDVT? I am not sure whether his UEDVT is associated with asymptomatic COVID19.
- I was wondering if you can provide some additional information, such as CBC, BMP, and other lab results.
- Regarding the treatment of recurrent DVT, more details for the difference between COVID positive or negative patients will be more helpful.
Round 2
Reviewer 1 Report
Thanks to the Authors for the effort in trying to answer to my comments. It seems to me that they were not fully involved in the therapeutic management of the patient, and it makes the difference, both for the past and for the current management. I totally disagree of the surgical management of the UEDVT in this patient, that in my opinion was fully unneeded. Moreover, again the reply to the comment n. 4 was not adequate because the Authors reported literature data about CDT or PMT...but my question was addressed to the patient: where are the US scan findings reporting the recanalization rate of the veins of the upper limb of this patient? Lastly, some flaws still are evident in the therapeutic management of the UEDVT as assessed by the International Guidelines
Reviewer 3 Report
The theme has clinical and scientific relevance. Overall, the article is clear and informative. I am ok with this manuscript for publication
This manuscript is a resubmission of an earlier submission. The following is a list of the peer review reports and author responses from that submission.
Round 1
Reviewer 1 Report
This is a case report concerning a recurrence of UEDVT secondary to COVID-19. In my opinion ther are some major concerns worthy to be discussed and clarified by the Authors.
- In the Introduction section is reported that the incidence of venous thromboembolic events has been noted in 20-55% of COVID patients. There are some papers reporting even higher rates, up to 75% in ICU setting and up to 30% in internal medicine/infective disease.
- Along the presentation of the case, it appears that the patient had atrial fibrillaton with an high risk profile (CHADS2VASc > 3) but he was not administered oral anticoagulants. Why? They were not contraindicated because he was treated with full-dose of apixaban for 3 months; so, why do not prescribe DOACs for his atrial fibrillation?
- Positioning of a subclavian PM represente a risk factor for developing an UEDVT. When diagnosed the index event, the patient received a 3-month course of oral anticoagulants (apixaban 5 mg twice-daily) only after 2 days of enoxaparin (which dose? it is not reported), avoiding the loading dose with apixaban with 10 mg twice-daily for 1 week. can the Authors clarify this therapeutic approach?
- Why only 3 months of anticoagulant therapy? Even if this was the first DVT experienced by the patient, the potential risk factor represented by the PM was not removed and, even better, it had to be kept in place. In this case, oral anticoagulant therapy had to be continued. Moreover, no data is reported about the recenalization rate of the subclavian vein: the presence of a residual thrombus mass, would have represented a further risk factor for a DVT recurrence.
- The Authors must consider the natural history of a DVT event and it is well known which is the recurrence rate of a UEDVT. But, despite the persistence of the PM and the natural history (not having information about the residual thrombus mass), when the recurrence of UEDVT occurred, again another 3-month course of anticoagulant therapy only was prescribed to the patient. Why this time was not prescribed a long-term course of DOACs?
- In my opinion, COVID19 played a secondary and minor, even irrelevant, role in this patient, because he was completely asymptomatic. The correlation between COVID19 and VTE is higher for patients admitted to hospital, and even higher when related to the setting (ICU more than medical wards). But there are no prospective data reported in the literature, about the VTE incidence in asymptomatic COVID19 patients quarantined at home.
Reviewer 2 Report
The described case is interesting, but it should not be published, especially in a journal with such a high IF. The patient's description is not carefully prepared, and four facts disqualify the manuscript as worthy of publication:
1. the patient was previously diagnosed with UEDVT, so what did the authors of the article expect this time?
2. the lack of a D-dimer result is a serious limitation.
3. UEDVT diagnosis is written very carelessly,
4. the uses of the therapy should be described more broadly - drug dosages, when drugs were administered - this is very important for subsequent clinicians who will be treating these patients.